# Comparative Nutritional Assessment and Metabolomics of a WRKY Rice Mutant with Enhanced Germination Rates

Santiago Bataller [1] , Anne J. Villacastin [1], Qingxi J. Shen [1] and Christine Bergman [2],*

[1] School of Life Sciences, University of Nevada, Las Vegas, 4505 S Maryland Parkway, Las Vegas, NV 89154, USA; santiago.bataller@unlv.edu (S.B.); ajvillacastin@lbl.gov (A.J.V.); jeffery.shen@unlv.edu (Q.J.S.)

[2] Food & Beverage and Event Management Department, University of Nevada, Las Vegas, 4505 S Maryland Parkway, Las Vegas, NV 89154, USA

* Correspondence: christine.bergman@unlv.edu

**Abstract:** Rice is the primary staple food for half the world's population. Climate change challenges and food insecurity supports the need for rice with agronomically advantageous traits. We report on a transposon insertional rice mutant with enhanced germination rates. This trait is advantageous for rice growth in limited water regions and to reduce yield constraints caused by weed and bird competition. Evaluations of vital nutritional components, compositional analysis, and comparative metabolomics on threshed grain samples are performed, as these assays are those used to assess the safety of foods from genetically modified crops. Compared with the wild type (cv. Nipponbare), *oswrky71* mutant grains have a similar size, shape, amount of crude fiber, crude fat, and ash content but higher crude protein. Mineral analyses reveal higher contents of phosphorus and zinc but lower calcium, potassium, sodium, and manganese in the mutant. Analysis of B vitamins reveals significantly higher riboflavin concentrations but lower choline chloride, calcium pantothenate, and thiamine. In addition, untargeted metabolomics analyses identify approximately 50 metabolites whose levels differed between the mutant and its wild type. Physical traits and compositional parameters analyzed are mostly similar and within the range or very close to being considered safe for consumption by the International Life Sciences Institute Crop Composition Database. Further agronomic evaluation and cooked rice sensory properties assessment are needed before positioning this mutant for human consumption.

**Keywords:** rice; germination; OsWRKY71; crop safety assessment; genetically modified; untargeted metabolomics; nutrition





## 1. Introduction

Rice is an essential source of kilocalories and various nutrients for more than a fifth of the world's population, critically highlighting its role in achieving food security. In the last harvest year, nearly 510 million metric tons of milled rice were produced worldwide, with about 80% allocated for human consumption [1]. Milled rice supplies considerable kilocalories and provides vital minerals such as magnesium, selenium, and phosphorus, as well as vitamins B1, B3, and B9 [2]. Whole-grain rice is a rich source of carbohydrates, dietary fiber, lipids, protein, phosphorus, potassium, B vitamins, and phytochemicals [3].

Challenges in rice crop production are vastly intensified as industrialization progresses. With the expansion of economies comes the loss of arable land, fewer labor resources, and less water access as urbanization persists and populations increase [4]. Along with these challenges are the adverse threats of climate change on rice yield and end-use quality. For example, fluctuating average temperatures negatively impact rice production [5]. Heat stress is predicted to negatively affect parameters such as germination rates, seedling growth, pollen grain number, and grain filling [6]. Currently, climate change seriously affects approximately 52% of the world's rice producers, including those in China, India,

and Indonesia, who provide about 75% of the total rice demand worldwide [7]. The aggravated rice production comes alongside increasing consumption and demand [8]. It is, therefore, critical that we find ways to ensure rice's stable and sustainable production despite these hurdles. Currently, the focus of cereal crop improvement programs includes the enhancement of seedling vigor, grain yield, and nutritional quality. These important traits and others are strongly controlled by genetic and environmental factors [9].

Pivotal to the cultivation of crops is the germination of seeds, which is the first step in the plant's life cycle. In contrast, seed dormancy is an evolutionarily important cereal trait that allows for the storage of seeds until conditions become optimal for germination and growth. The ability of seeds to germinate or remain dormant is still a poorly understood area of research despite some significant breakthroughs in the field [10]. Seed dormancy entails a complex balance of molecular, biochemical, and physiological processes controlled by coordinating intrinsic and extrinsic cues [11]. Among these factors, the intricate crosstalk between two phytohormones—gibberellic acid (GA) and abscisic acid (ABA)—have long been shown to be primary endogenous players regulating the ability of seeds to germinate or stay dormant [12]. ABA promotes dormancy of seeds as it inhibits germination, while in contrast, the GA breaks this dormancy and promotes germination.

Pre-harvest sprouting of seeds, while the kernels are still attached to the plant, leads to a severe loss of quantity and quality of cereal harvest, a phenomenon especially significant in humid regions [11,13]. Additionally, pre-harvest sprouting threatens the plant's growth as the lower grain weight is prone to infestation of saprophytic fungi [14]. On the other hand, prolonged dormancy of seeds also creates challenges, especially for farmers who plant via direct dry seeding [15–18]. Under this type of planting, prolonged seed germination times require fields to be flooded for longer periods of time. Shortening the time rice is ponded reduces water use by limiting evaporation and deep drainage below the crop root zone [15]. This is especially important in water-constrained regions. In addition, the longer it takes for seeds after planting to become seedlings, the greater the potential for yield loss caused by bird consumption of the seeds and for weeds to outcompete the rice seedlings, especially under organic production methods [16,17]. To date, research designed to improve rice germination rates has been limited [18,19]. Varieties that germinate rapidly and uniformly would be a valuable contribution to global rice-improvement efforts.

We have identified a rice mutant line from a rice transposon insertion library [20] that exhibited an early germination phenotype because of the knockout of a specific gene, *OsWRKY71*, a member of the WRKY superfamily transcription factors. WRKY transcription factors are highly conserved in the plant kingdom, and some members are key players in growth, development, and stress response [21]. An invariant amino acid sequence characterizes WRKYs, WRKYGQK, followed by a zinc-finger binding domain, both of which allow for the WRKY's specificity for gene targets that have *cis*-regulatory elements called the W-box [22]. OsWRKY71 has been previously shown to be involved in the complex interaction of GA and ABA during germination [23–25]. In previous studies on rice aleurone cells, OsWRKY71 was shown to be induced by ABA, and its protein product binds to the W-box of GA-induced α-amylase genes [24,25]. It was proposed that the binding prevented the expression of the amylase and other GA-responsive genes and hindered germination. Our recent studies demonstrated that knocking out the gene led to a faster germination rate, suggesting this gene may be an important target in our efforts to find crops with a more controlled seed-sprouting phenotype. We believe this mutant line may be essential as we continue to find solutions to improve crop productivity and quality.

In this current study, we assess the nutritional properties of the *oswrky71* rice mutant. We also used metabolomics to analyze the global metabolite differences between the wildtype and mutant kernels to evaluate the effect of the mutation comprehensively. As we recognize the agronomic value of this mutant's early germination phenotype, this study aims to provide an exhaustive compositional and nutritional evaluation relative to its wild-type counterpart as a key step in the risk assessment process before its distribution for human consumption.

## 2. Materials and Methods

### 2.1. Plant Materials and Growth Conditions

*Oryza sativa* ssp. *japonica* cv. Nipponbare dSpm mutants were acquired from the Sundaresan Lab, University of California, Davis, CA, USA [21]. Wild-type (c.v. Nipponbare) and mutant seeds were sterilized with 10% bleach (The Clorox Company, Oakland, CA, USA) for 20 min and were then washed five times with sterile Milli-Q water for 2 min per wash. Seeds were placed on sterile filter paper in a Petri dish with sterile Milli-Q water, and germination occurred in a dark incubator at 28 °C. After 14 days, seedlings were transferred to a greenhouse at the University of Nevada, Las Vegas and transplanted to 3:1 soil blend of Turface MVP calcined clay (Turface, Buffalo Grove, IL, USA) and lawn soil (Sta-Green, Mechanicsville, VA, USA). Plants were flooded every 6 h with water supplemented with Miracle-Gro All Purpose Plant Food (Scotts, Marysville, OH, USA). Growth conditions were maintained at 50–60% humidity, $28 \pm 5$ °C, with shaded natural lighting. Plants were grown to maturity in 6 months in the greenhouse, after which panicles were harvested. Nipponbare grows to maturity in approximately 4 months under field conditions [26]. However, the seeds were ready for harvest in the greenhouse after 6 months. Seed dormancy was broken by storing the seeds in the dark at room temperature for at least one month before germination assays.

### 2.2. Genotyping Mutant Lines by PCR

To verify homozygous mutant lines, genomic DNA was extracted from leaf tissues, and PCR was performed to detect for insertion of the dSpm transposon. Approximately 2 cm of leaf blade tissues were collected for each line and homogenized using the MP FastPrep-24 homogenizer (MP Biomedicals, Irvine, CA, USA). Then, 500 µL Edwards buffer (0.2M Tris-HCl pH 7.5, 0.25M NaCl, 0.025M EDTA pH 8.0, 0.5% *w/v* SDS) was added into each sample, and cell debris was separated out by centrifugation ($21,000 \times g$) for five minutes. Then, 230 µL isopropanol was added to 300 µL of supernatant, and DNA was pelleted out by centrifugation ($21,000 \times g$) for 10 min. The DNA pellet was washed thrice with 70% ethanol by centrifugation ($21,000 \times g$) for five minutes. Extracted genomic DNA served as a template for PCR diagnostics using GoTaq Green Master Mix (Promega, Fitchburg, WI, USA) and oligonucleotides listed in Supplementary Table S1. The following cycling conditions were used: initial denaturation at 95 °C, followed by 30 cycles of 95 °C denaturation, 60 °C annealing and 72 °C extension, and final extension at 72 °C for five minutes. The products were analyzed on 1.0% agarose gel stained with ethidium bromide.

### 2.3. Germination Assays

For germination assays, whole seeds were sterilized as described previously. All germination assays occurred in a dark 28 °C incubator, using at least 25 seeds per replicate and at least three replicates. For liquid germination assays, 15 mL of sterile Milli-Q water was used in a 50 mm Petri dish. Solid germination assays occurred on $1/2\times$ Murashige and Skoog Media (pH 5.8) supplemented with 1% (*w/v*) agar and 0.05% (*w/v*) MES. Germination was determined by a radicle or coleoptile growth of at least 2 mm. Statistical significance in germination rate between the wild type and *oswrky71* at each time point was determined via unpaired Student's *t*-tests. T50 calculations were performed as previously described [27], and statistical significance was determined via Tukey's honestly significant difference test.

### 2.4. Kernel Morphology

Phenotypic analyses were performed following the Standard Evaluation System for Rice (SES) [28] methods with minor modifications. Mature seeds and kernels were scanned using a Canon 8800F Scanner, and ImageJ was used to measure the length, width, and area [29]. For the 100-grain weight, 100 well-developed whole grains dried to less than 13% moisture content were weighed on a Mettler Ae50 analytical balance in triplicates. Rice

grains were threshed using a compact rice husker (TR-200, Kett, Tokyo, Japan). For alkali digestion, four threshed-rice kernels were placed into a Petri dish with 10 mL 1.7% KOH and were arranged to prevent touching. After incubating for 23 h at 30 °C, the kernels were scored for spreading following the criteria in SES. Threshed rice samples were used for the subsequent analyses (proximate composition, total protein, amino acids, B vitamins, and metabolomics).

### 2.5. Proximate Composition

The proximate composition analyses (i.e., crude fiber, crude protein, crude fat, ash, and carbohydrate by difference) were performed using the official analysis methods by the Association of Official Agricultural Chemists (AOAC). After harvest, whole grains were dried to a final moisture content of 10 to 12%. Threshed samples were ground into flour before evaluation. Analyses were performed in triplicates, and data were expressed on a dry weight basis (DB).

### 2.6. Total Protein

The protein concentration of threshed, dry seeds was determined using the Pierce quantitative protein assay kit (Thermo Fisher Scientific, Waltham, MA, USA) following the manufacturer's protocol. A standard curve was generated using the manufacturer's procedure and the Peptide Digest Assay Standard. Threshed kernels were ground into flour using liquid nitrogen, and 0.095 g of each sample was used for protein extraction. The working reagent (WR) was prepared by mixing 50 parts of colorimetric peptide assay reagent A, 48 parts of colorimetric assay reagent B, and 2 parts of colorimetric assay reagent C. The WR was kept at room temperature for less than 30 min. Then, 20 μL standard or sample extract was pipetted into a well of a 96-well microplate. To each well, 180 μL of the WR was added, and the plates were thoroughly mixed on a plate shaker for 30 s to 1 min. The plate was covered and incubated at 37 °C for 30 min. Absorbance was measured at 562 nm.

### 2.7. Amino Acids

Amino acids quantification of threshed, dry seeds was performed using three replicates for each genotype. Samples were ground in liquid nitrogen. Then, 6 N HCl was added to each sample and incubated at 110 °C to facilitate digestion. After 21 h of incubation, a 50 μL supernatant was taken, and a 4 N NaOH and 50 μL deproteinization reagent were added. After mixing with a vortex, the mixture was centrifuged at 13,200 rpm for 4 min. A total of 8 μL was transferred into a tube and mixed with 42 μL of the buffer. Afterward, 20 μL derivation reagent was added. The mixture was incubated for 15 min at 55 °C, and 50 μL of the mixture was used for downstream analysis. Amino acids were quantified using a Shimadzu LC20AD-API 3200MD TRAP HPLC-MS/MS.

### 2.8. B Vitamins

B-vitamin levels of threshed, dry seeds were determined using LC-MS/MS measurements on an Agilent 1290 UHPLC system coupled to an Agilent 6470 triple quadrupole system. System suitability tests were measured along with the samples to verify the calibration. The threshed samples were ground, and the vitamins were extracted from the resulting flour. Samples of 0.5 g were used for extraction in 5 mL of 50% methanol in water. Analyses were performed in triplicate.

### 2.9. Untargeted Metabolomics

Metabolomics was performed using 50 mg of each sample. The samples were precisely weighed into a 2 mL Eppendorf tube and extracted with 800 μL 80% methanol, and all samples were homogenized on a MM 400 mill mixer at 60 Hz for 2 min, followed by sonication for 30 min, 4 °C. Then, all samples were kept at −40 °C for 1 h. After that, samples were vortexed for 30 s and centrifuged at 12,000 rpm at 4 °C for 15 min. A 200 μL supernatant

and 5 μL of DL-o-Chlorophenylalanine (140 μg/mL) were transferred to a vial for LC-MS analysis. Separation was performed by Ultimate 3000LC combined with Q Exactive MS (Thermo) and screened with ESI-MS (targeted MS/MS mode). The LC system comprised a Thermo Hyper Gold C18 (100 × 2.1 mm, 1.9 μm) with Ultimate 3000LC. The mobile phase was composed of solvent A (0.1% formic acid-5% acetonitrile-water) and solvent B (0.1% formic acid-acetonitrile) with gradient elution (0–1.5 min, 100–80% A; 1.5–9.5 min, 80–0% A; 9.5–14.5 min, 0% A; 14.5–14.6 min, 0–100% A; 14.6–18.0 min, 100% A). The flow rate of the mobile phase was 0.3 mL·min$^{-1}$. The column temperature was maintained at 40 °C, and the sample manager temperature was set at 4 °C. Mass spectrometry parameters in ESI+ were as follows: heater temp 300 °C; sheath gas flow rate, 45 arb; aux gas flow rate, 15 arb; sweep gas flow rate, 1 arb; spray voltage, 3.0 kV; capillary temp, 350 °C; S-Lens RF level, 30%. Mass spectrometry parameters in ESI− were as follows: heater temp 300 °C; sheath gas flow rate, 45 arb; aux gas flow rate, 15 arb; sweep gas flow rate, 1 arb; spray voltage, 3.2 kV; capillary temp, 350 °C; S-Lens RF level, 60%. At the beginning of the sequence, we ran three quality control (QC) samples to avoid small changes in chromatographic retention time and signal intensity. Quality control (QC) samples were used to evaluate the methodology. The exact amount of extract was obtained from each sample and mixed as QC samples. The QC sample was prepared using the same sample preparation procedure. Using the data of accurate masses and MS/MS fragments, the chemical structures of important metabolites were identified using the following online databases: Human Metabolome Database (www.hmdb.ca; accessed on 29 September 2022), Metlin (www.metlin.scripps.edu; accessed on 29 September 2022), and the Mass Bank (www.massbank.jp; accessed on 29 September 2022) [30–32].

### 2.10. Statistical Methods

All samples were analyzed statistically using the Student's *t*-test. For untargeted metabolomics, the raw data were acquired and aligned using Compound Discoverer (3.0, Thermo) based on the *m/z* value and the retention time of the ion signals. For multivariate analysis, ions from ESI− and ESI+ were merged and imported into the SIMCA-P program (version 14.1). A principal components analysis (PCA) was first used as an unsupervised method for data visualization and outlier identification. Supervised regression modeling was then performed on the data set using partial least squares discriminant analysis (PLS-DA) or orthogonal partial least squares discriminant analysis (OPLS-DA) to identify the potential biomarkers. The biomarkers were filtered and confirmed by combining the results of the VIP values (VIP > 1.5) and *t*-test ($p < 0.05$).

## 3. Results

### 3.1. Transposon-Mediated Mutation in OsWRKY71 Results in Early Germination Phenotype

Previous transient expression studies in barley aleurone showed that OsWRKY71 is a transcriptional repressor of the GA-induced α-amylase gene, Amy32b [24,25]. Hydrolytic enzymes such as α-amylases play an essential role in the mobilization of stored reserves during germination, providing nutrients to the growing embryo [33–35]. So, we hypothesized that a loss-of-function mutation in *OsWRKY71* would result in an enhanced germination rate. To investigate the role of OsWRKY71 in seed germination, we obtained a rice mutant, *oswrky71* [20]. Sequencing confirmed that the stable dSpm transposon insertions in *oswrky71* occur in the 5′UTR (27 bp downstream of transcriptional start site (TSS) (Figure 1A). A PCR-driven breeding strategy was performed to ensure homozygosity of the *oswrky71* mutation (Supplementary Table S1).

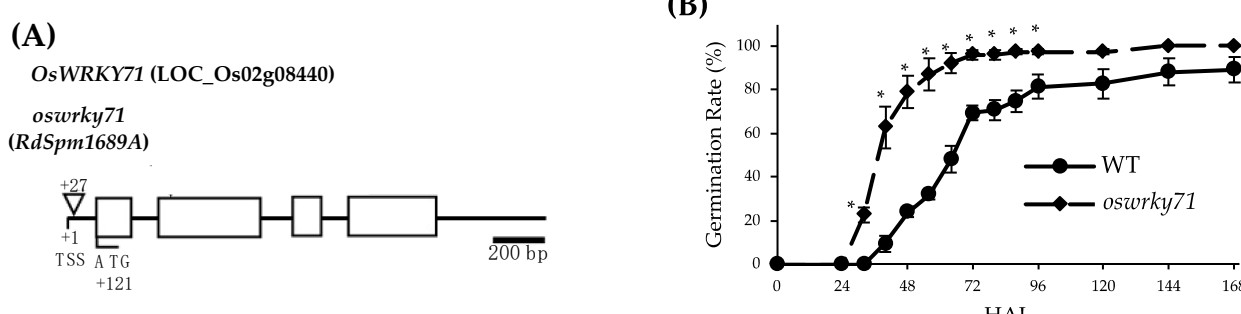

**Figure 1.** Mutant *oswrky71* seeds display an early germination phenotype. (**A**) Gene diagram for *OsWRKY71* showing the position of dSpm transposon insertions in *oswrky71*. Transcriptional start site (TSS) and start codon (ATG) are marked. Boxes indicate exons. Triangles with arrows represent the position of each insert. The number of base pairs downstream from TSS is shown. Scale is 200 bp. (**B**) Quantification of germination rates for wild-type (WT) and *oswrky71* mutant. Seeds were incubated at 28 °C in the dark on ½× MS plates (0.05% MES (*w*/*v*), pH = 5.8). Germination was defined as a radicle or coleoptile reaching approximately 2 mm in length. Error bars represent ± SEM. Asterisks indicate significance for one mutant relative to the wild type in Student's *t* test ($p < 0.05$). $n = 3$ biological replicates, each with 25 seeds.

To determine whether the loss of a negative regulator of GA signaling could affect seed germination rates, *oswrky71* and wild-type seeds were subjected to dark-grown germination assays in water. Seed germination rates were measured every 8 h, starting after 24 h, with germination determined by at least 2 mm of radicle or coleoptile growth. Relative to the wild type, *oswrky71* mutants displayed a 1.4–1.8-fold increased germination rates at 36 h after imbibition (HAI) and a 1.3-fold increase at 48 HAI (Figure 1B). No significant differences were detected after 60 HAI when all lines approached maximum germination. The time to reach a 50% maximal germination (T50) was determined using the Farooq method [36], with *oswrky71* mutant seeds reaching T50 at 38.0 HAI, which was 17.7% earlier than the 46.1 HAI observed for the wild type. Thus, the *oswrky71* mutant displayed an early seed-germination phenotype.

### 3.2. Morphological Characterization of oswrky71 Kernel

To determine the effects of mutating *OsWRKY71* on grain morphology, we used the standard evaluation system for rice [28]. Mutant kernels have similar seed surface area, length, and width compared to the wild type (Figure 2B,C). For the overall shape, kernels of both genotypes were categorized as 'short' (length < 5.5 mm) and 'bold' (length/width ratio = 1.1–2.0 mm). No difference in seed coat color (i.e., light brown) was observed between the wild type and mutants. In addition, mutant lines exhibited significantly lower 100-grain weight ($p < 0.05$; Figure 2D). Alkali spreading value (ASV), an indicator of gelatinization temperature (GT), was performed to assess this aspect of the grain end-use quality [28,37]. ASV is inversely related to GT, the temperature at which starch granules melt and the grains start to cook. It positively correlates with cooking time and texture in domesticated rice [38]. Wild type and *oswrky71* had a mean ASV value of approximately 4, corresponding to intermediate GT for both genotypes (Figure 2E) [28].

### 3.3. Proximate Composition and Mineral Analyses

The proximate compositions of the wild-type and mutant kernels according to the Association of Official Agricultural Chemists' (AOAC) official methods of analysis are shown in Table 1, which was calculated on a dry weight basis except for moisture, which is expressed on a fresh weight basis, to allow comparison with each other. The moisture content (MC) of both wild-type (11.6% MC) and mutant (9.6% MC) kernels is lower than the safe moisture content required for secure storage of processed rice during short periods (14% MC) and long-term storage (12% MC) to avoid microbial contamination and insect

infestation [39–41]. Crude fat values were not significantly different between the wild type (1.6%) and mutant (1.7%). The crude protein content of the mutant line was significantly higher (13.7%) compared to the wild type (13.1%) ($p < 0.05$). The ash content values were similar between the wild type (4.1%) and the mutant (3.7%). The fiber percentages of the wild type (9.8%) and the mutant (9.8%) were the same. The total and available carbohydrates in 100 g of dehulled grains calculated by difference were similar between the wild type (69.7 g and 73.7 g, respectively) and the mutant (78.3 g and 82.4 g, respectively).

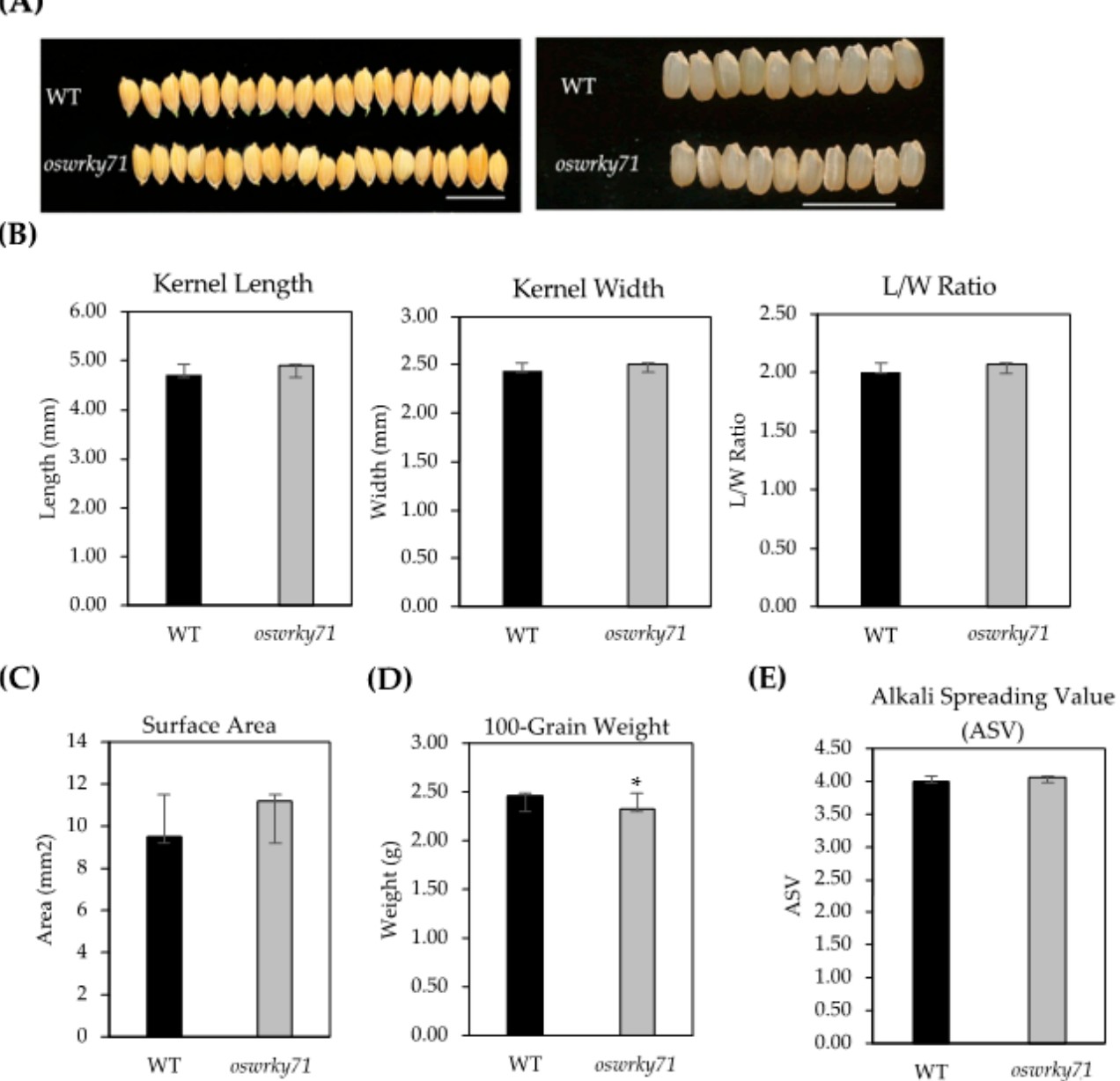

**Figure 2.** Seed morphology of *oswrky71*. (**A**) Representative images of WT and *oswrky71* mutant seeds. Scale bar = 1 cm. (**B**) Length, width, length–width (L/W) ratio, and (**C**) surface area of WT and *oswrky71* mutant kernels. Error bars represent ± SD. *n* = 100 kernels. (**D**) 100-grain weight of WT and *oswrky71* kernels. Error bars represent ± SD. *n* = 4 biological replicates, each with 100 kernels. (**E**) alkali spreading value (ASV) of WT and *oswrky71*. Error bars represent ± SD. Asterisks indicate significance for one mutant relative to the wild type in Student's *t*-test ($p < 0.05$). *n* = 4 biological replicates, each with 4 kernels.

**Table 1.** Comparison of the proximate composition of threshed rice grain.

| Component | WT | | *oswrky71* | | *p*-Value [b] | Lit. Range [c] |
|---|---|---|---|---|---|---|
| | Mean ± SD [a] | Range | Mean ± SD [a] | Range | | |
| Crude protein (% DB) | 13.10 ± 0.2 | 12.9–13.2 | 13.70 ± 0.2 | 13.6–13.9 | 0.01 | 5.9–11.8 |
| True protein (mg) | 13.03 ± 1.8 | 11.1–14.7 | 9.49 ± 1.3 | 8.7–11.0 | 0.05 | Not available |
| Crude fiber (% DB) | 9.77 ± 0.3 | 9.5–10.1 | 9.83 ± 0.4 | 9.5–10.3 | 0.83 | 8.6–18.2 |
| Crude fat (% DB) | 1.60 ± 0.0 | 1.6–1.6 | 1.70 ± 0.1 | 1.6–1.8 | 0.16 | 0.56–3.47 |
| Ash (% DB) | 4.07 ± 0.1 | 4.0–4.1 | 3.70 ± 0.3 | 3.4–3.9 | 0.08 | 3.61–8.6 |
| Carbohydrate (% DB) | 81.23 ± 0.2 | 81.1–81.4 | 80.83 ± 0.4 | 80.5–81.3 | 0.19 | 80.0–86.4 |
| Moisture (% FW) | 11.57 ± 0.3 | 11.4–11.9 | 9.57 ± 0.86 | 8.8–10.5 | 0.02 | 3.5–73.7 |

[a] Values are means ± SD (*n* = 3). Values are expressed on a dry basis (DB) except for moisture, which is expressed on a fresh weight basis (FW). [b] Statistical significance was analyzed using Student's *t*-test ($p < 0.05$). [c] Combined reference and ILSI-CCDB range.

Mineral contents were analyzed for the wild type and the mutant kernels on a dry weight basis and are shown in Table 2. Mineral analyses revealed a higher content of phosphorus and zinc in the mutant grains but lower calcium, potassium, sodium, and manganese ($p < 0.05$). Magnesium, sulfur, iron, copper, and selenium contents were similar for both genotypes.

**Table 2.** Comparison of the mineral composition of threshed rice grain.

| Mineral (mg/100 g DB) | WT | | *oswrky71* | | *p*-Value [b] | Lit. Range [c] |
|---|---|---|---|---|---|---|
| | Mean ± SD [a] | Range | Mean ± SD [a] | Range | | |
| Calcium | 70.00 ± 0.0 | 70–70 | 50.00 ± 0.0 | 50–50 | 0.00 | 10–150 |
| Phosphorus | 380.00 ± 0.0 | 380–380 | 416.67 ± 5.8 | 410–420 | 0.00 | 190–470 |
| Magnesium | 160.00 ± 10.0 | 150–170 | 180.00 ± 10.0 | 170–190 | 0.07 | 30–170 |
| Potassium | 706.67 ± 11.5 | 700–720 | 640.00 ± 26.5 | 610–660 | 0.02 | 170–472 |
| Sulfur | 146.67 ± 1.5 | 140–150 | 143.33 ± 5.8 | 140–150 | 0.52 | 30–220 |
| Sodium | 33.67 ± 1.5 | 32–35 | 10.00 ± 0.0 | 10–10 | 0.00 | 0–100 |
| Zinc | 4.17 ± 0.2 | 4.2–4.0 | 6.0 ± 0.1 | 5.9–6.1 | 0.00 | 2.2–5.3 |
| Iron | 8.63 ± 1.4 | 7.7–10.2 | 7.0 ± 0.2 | 6.8–7.1 | 0.11 | 1.6–9.08 |
| Manganese | 6.67 ± 0.2 | 6.5–6.8 | 5.47 ± 0.2 | 5.3–5.7 | 0.00 | 2–11.7 |
| Copper | 1.30 ± 0.1 | 1.2–1.4 | 1.33 ± 0.1 | 1.3–1.4 | 0.64 | 0.2–1.3 |

[a] Values are means ± SD (*n* = 3). Values are expressed on a dry basis (DB). [b] Statistical significance was analyzed using Student's *t*-test ($p < 0.05$). [c] Combined reference and ILSI-CCDB range.

### 3.4. Amino Acids Composition Analysis

A comparison of the amino acid composition of the wild-type and the mutant kernels is shown in Table 3. There were no statistically significant differences in concentrations of most amino acids except for the reduced valine, aspartic acid, lysine, and phenylalanine concentrations in the mutant kernels ($p < 0.05$). The total protein (true protein) was also reduced in the mutant kernel (Table 1).

### 3.5. Quantification of B Vitamins

To determine the concentrations of water-soluble B vitamins (choline chloride, thiamine, calcium pantothenate, and riboflavin) in wild-type and mutant kernels, liquid chromatography with tandem mass spectrometry (LC-MS/MS) was performed. Analysis of B vitamins showed significantly higher riboflavin concentrations in the mutant kernels but lower choline chloride, calcium pantothenate, and thiamine (Table 4).

### 3.6. Identification of Differential Metabolites

To compare the metabolite profiles between the wild-type and the mutant kernels, an untargeted metabolomic evaluation was performed using ultra-performance liquid chromatography time of flight mass spectroscopy (UPLC-TOF-MS) (Figure 3A). To demonstrate the stability of the LC-MS system, quality control (QC) samples were used in both positive and negative electrospray ionization (ESI) modes. The relative standard deviation



(%RSD) of the QC samples was less than 30%, suggesting that the analysis procedure was robust and can be used for subsequent sample analysis. Orthogonal projections to latent structures-discriminant analysis (OPLS-DA) for both positive and negative modes showed clear separations between the wild-type and mutant metabolic profiles, indicating significant differences in metabolites.

**Table 3.** Comparison of the amino acid composition of threshed rice grain.

| Amino Acid (mg/100 g DB) | WT | | *oswrky71* | | *p*–Value [b] | Lit. Range [c] |
|---|---|---|---|---|---|---|
| | Mean ± SD [a] | Range | Mean ± SD [a] | Range | | |
| Glycine | 462.7 ± 21.5 | 443.2–485.8 | 389.0 ± 48.7 | 360.8–445.2 | 0.07 | 290–510 |
| Alanine | 695.7 ± 7.0 | 688.1–701.8 | 622.2 ± 60.5 | 554.6–671.5 | 0.10 | 330–630 |
| Serine | 497.4 ± 19.9 | 481.8–519.8 | 425.2 ± 43.6 | 374.9–450.6 | 0.06 | 230–560 |
| Proline | 524.8 ± 14.9 | 510.2–540.0 | 469.5 ± 44.7 | 417.9–496.8 | 0.11 | 280–540 |
| Valine | 730.0 ± 38.3 | 700.5–773.3 | 612.8 ± 59.9 | 550.0–669.3 | 0.05 | 340–650 |
| Threonine | 382.6 ± 5.7 | 376.5–387.8 | 334.0 ± 42.1 | 285.8–364.1 | 0.12 | 220–410 |
| Cysteine | 125.0 ± 23.4 | 104.4–150.5 | 125.2 ± 16.3 | 114.8–143.9 | 0.99 | 100–260 |
| Isoleucine | 439.6 ± 10.4 | 427.9–447.3 | 378.3 ± 44.5 | 328.3–413.7 | 0.08 | 240–460 |
| Aspartic acid | 1335.8 ± 39.2 | 1301.9–1378.8 | 1143.0 ± 81.5 | 1050.2–1203.1 | 0.02 | 500–990 |
| Glutamic acid | 2752.6 ± 72.0 | 2676.5–2819.7 | 2341.1 ± 291.9 | 2054.0–2647.4 | 0.08 | 890–1990 |
| Methionine | 91.8 ± 59.2 | 24.4–135.1 | 93.9 ± 4.2 | 89.3–97.4 | 0.95 | 130–310 |
| Histidine | 222.6 ± 9.0 | 212.9–230.7 | 197.8 ± 23.5 | 171.7–217.3 | 0.16 | 140–281 |
| Phenylalanine | 499.5 ± 7.2 | 491.1–504.1 | 434.1 ± 29.4 | 401.4–458.5 | 0.02 | 280–620 |
| Arginine | 920.8 ± 105.6 | 830.8–1037.0 | 775.9 ± 51.9 | 717.0–814.6 | 0.10 | 410–850 |
| Tryptophan | 4.7 ± 0.1 | 4.6–4.7 | 5.9 ± 1.7 | 4.8–7.8 | 0.28 | 50–180 |
| Lysine | 736.2 ± 34.1 | 698.7–765.4 | 580.3 ± 72.8 | 526.2–663.1 | 0.03 | 210–430 |
| Tyrosine | 321.5 ± 22.2 | 301.4–345.3 | 279.9 ± 19.5 | 260.5–299.4 | 0.07 | 50–180 |
| Leucine | 937.1 ± 25.1 | 919.9–965.9 | 814.7 ± 80.6 | 727.3–886.1 | 0.07 | 460–920 |

[a] Values are means ± SD (*n* = 3). Values are expressed on a dry basis (DB). [b] Statistical significance was analyzed using Student's *t*-test (*p* < 0.05). [c] Combined reference and ILSI-CCDB range.

**Table 4.** Comparison of the B vitamin composition of threshed rice grain.

| Vitamin (mg/kg DB) | WT | | *oswrky71* | | *p*-Value [b] | Lit. Range [c] |
|---|---|---|---|---|---|---|
| | Mean ± SD [a] | Range | Mean ± SD [a] | Range | | |
| Choline chloride | 555.33 ± 21.46 | 541–580 | 448.33 ± 23.71 | 423–470 | 0.00 | Not available |
| Calcium pantotenate (Vit B5) | 4.88 ± 0.09 | 4.88–4.96 | 3.65 ± 0.03 | 3.61–3.66 | 0.00 | 7.22–14.0 |
| Riboflavin (Vit B2) | 0.33 ± 0.01 | 0.32–0.34 | 0.36 ± 0.01 | 0.35–0.36 | 0.02 | 0.4–1.4 |
| Thiamine (Vit B1) | 2.72 ± 0.02 | 2.69–2.74 | 2.47 ± 0.03 | 2.44–2.50 | 0.00 | 2.35–6.25 |

[a] Values are means ± SD (*n* = 3). Values are expressed on a dry basis (DB). [b] Statistical significance was analyzed using Student's *t*-test (*p* < 0.05). [c] Combined reference and ILSI-CCDB range.

To identify the differential metabolites in the mutant kernels, the significantly changed metabolites between the two groups are filtered out based on a variable importance-in-projection (VIP) score (VIP > 1.5) for both positive and negative ESI modes (Figure 3B,C). Metabolites with VIP > 1.5, |log2FC| > 1.0, and *p*-value < 0.05 were considered differential metabolites (Figure 3C,D). Out of 3425 metabolites identified in the positive mode, 43 were differential metabolites, of which 41 were upregulated (log2FC > 1) and 2 were downregulated (log2FC < −1) in the mutant kernels (Supplementary Tables S2 and S3). In the negative mode, out of 2605 identified, 31 were differential metabolites (28 upregulated, 3 downregulated; Supplementary Tables S4 and S5). Furthermore, 11 differential metabolites appeared in both positive and negative ESI modes, which are significantly increased in the mutant kernel (Figure 4A,B). The differential metabolites identified can be categorized into eight super-classes [30]: lipids and lipid-like molecules, organic acids and derivatives, organic nitrogen compounds, organic oxygen compounds, organoheterocyclic compounds, phenylpropanoids and polyketides, alkaloids and derivatives, and benzenoids. Hierarchical cluster analysis of the differential metabolites shows that mutation in *OsWRKY71* led to more upregulated than downregulated metabolites in the kernels (Figure 4A,B). For positive and negative ESI modes, most differential metabolites belong to

the major class of metabolites, carboxylic acids, which include amino acids, peptides, and analogs [30]. The next major class of metabolites that were increased were the organooxygen compounds. One of the most important compounds in this class is pantothenic acid, also called vitamin B5, which is significantly upregulated in the mutant kernel. Another upregulated metabolite of known nutritional importance is L-carnitine, an organonitrogen compound [42,43]. There were only five significantly downregulated metabolites: corchorifatty acid F, propyl gallate, N-(1-Deoxy-1-fructosyl)tyrosine, suberic acid, and niazirinin. All these differentially expressed metabolites are present in embryo and endosperm [44].

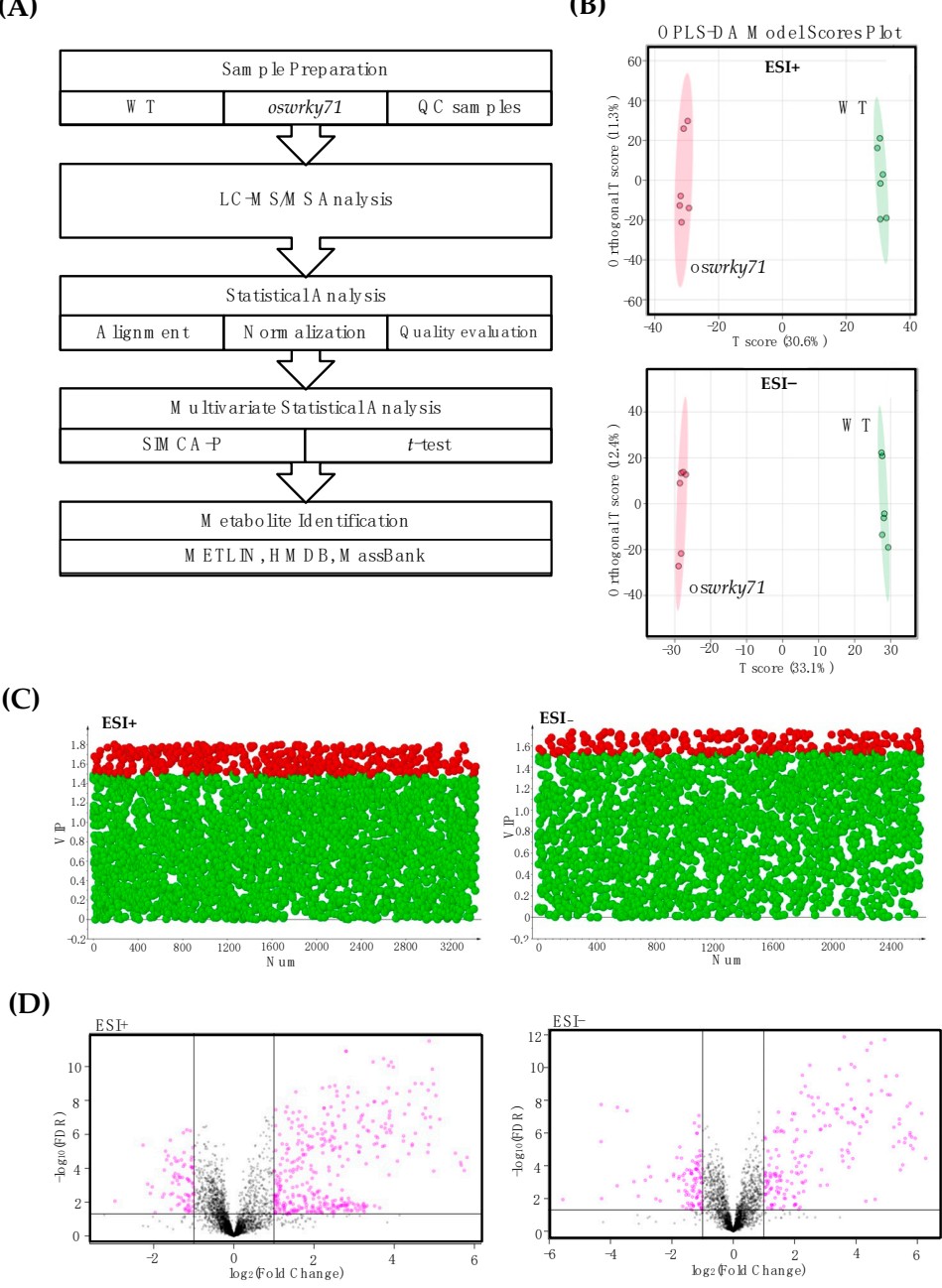

**Figure 3.** Untargeted metabolomics between WT and *oswrky71* threshed grain samples. (**A**) Flowchart showing a pathway for untargeted metabolomics and subsequent bioinformatic analysis. (**B**) Scores scatter plot of the OPLS-DA model in both positive (ESI+) and negative (ESI−) ion modes. (**C**) Distribution of VIP values (VIP > 1.5) in both positive (ESI+) and negative (ESI−) ion modes. Red dots indicate significant metabolites. (**D**) Volcano plots of differentially expressed metabolites (purple dots) in *oswrky71* threshed grain samples in both positive (ESI+) and negative (ESI−) ion modes.

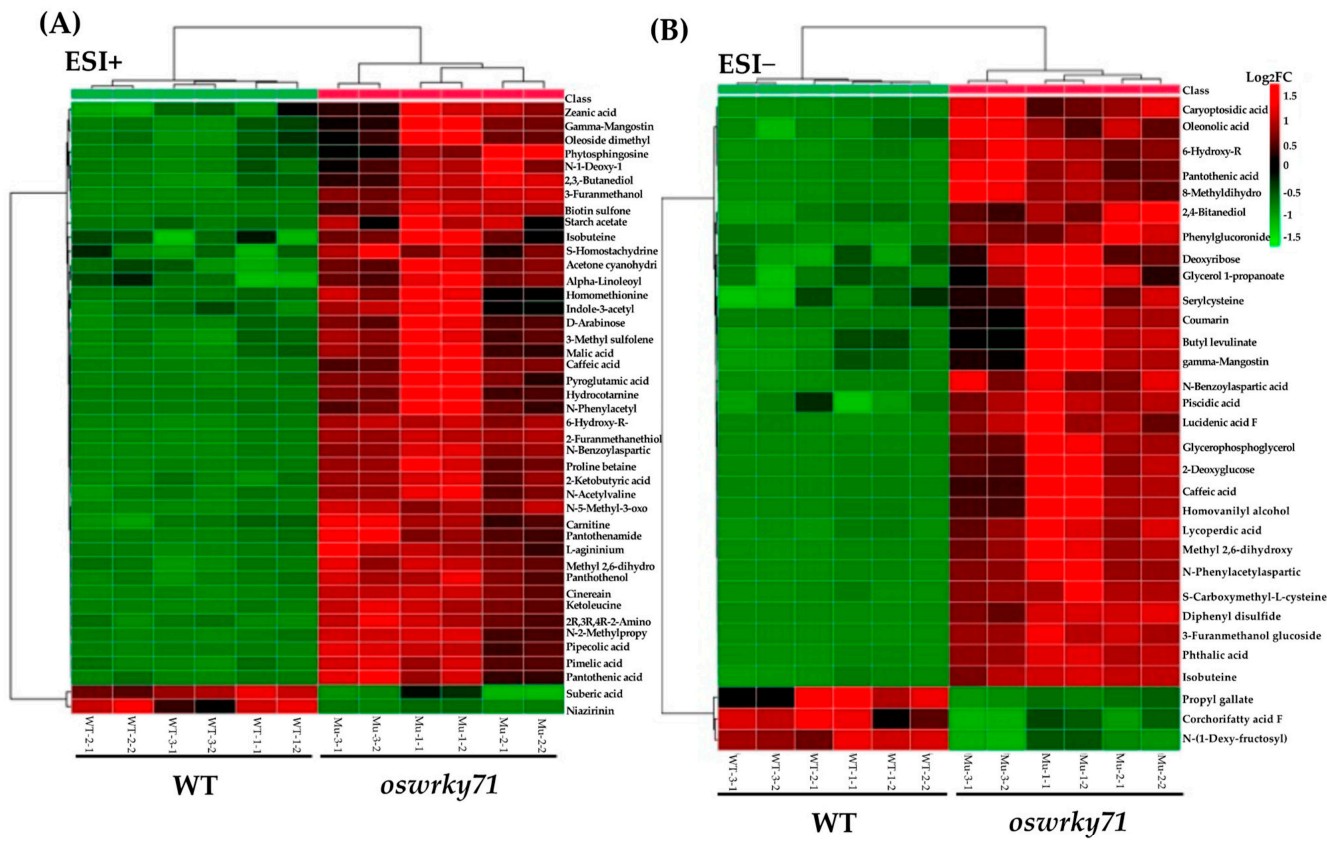

**Figure 4.** Heat map showed the differential metabolites between WT and *oswrky71* threshed grain samples. (**A**) Hierarchical cluster analysis of differential metabolites from ESI+ mode. (**B**) Hierarchical cluster analysis of differential metabolites from ESI− mode. Red indicates upregulation of metabolite expression; green, downregulation.

To investigate the metabolic pathways perturbed by the mutation in *OsWRKY71*, we performed pathway enrichment analysis using KEGG and MBRole [45,46]. All significant metabolites (Supplementary Tables S2 and S4) are imported to MBRole to obtain the categorical annotations, including pathways, enzyme interactions, and other biological annotations. There are mainly 34 and 51 enriched metabolic pathways from the differential metabolites in positive and negative ESI modes, respectively, that were affected by the mutation of *OsWRKY71* (Tables 5 and 6). Specifically, in the positive mode, aminoacyl-tRNA biosynthesis, amino acid metabolism, biosynthesis of plant hormones, and biosynthesis of alkaloids were among the pathways enriched. Biosynthesis of unsaturated fatty acids, amino acid metabolism, biosynthesis of phenylpropanoids, and alkaloids were a few of the enriched pathways from differential metabolites in the negative ESI mode.

**Table 5.** Pathway enrichment analysis of differential metabolites in ESI+ scan.

| Pathway | *p*-Value | FDR |
|---|---|---|
| Metabolic pathways | $7.7 \times 10^{-13}$ | $8.4 \times 10^{-11}$ |
| Aminoacyl-tRNA biosynthesis | $4.1 \times 10^{-11}$ | $2.2 \times 10^{-9}$ |
| Glycine, serine and threonine metabolism | $5.3 \times 10^{-9}$ | $1.9 \times 10^{-7}$ |
| Cysteine and methionine metabolism | $2.2 \times 10^{-8}$ | $6.5 \times 10^{-7}$ |
| ABC transporters | $3.6 \times 10^{-6}$ | $7.8 \times 10^{-5}$ |
| Alanine, aspartate and glutamate metabolism | $10.0 \times 10^{-6}$ | $1.8 \times 10^{-4}$ |

**Table 5.** *Cont.*

| Pathway | *p*-Value | FDR |
|---|---|---|
| Biosynthesis of plant hormones | $1.5 \times 10^{-5}$ | $2.3 \times 10^{-4}$ |
| Glucosinolate biosynthesis | $3.0 \times 10^{-5}$ | $4.1 \times 10^{-4}$ |
| Histidine metabolism | $4.2 \times 10^{-5}$ | $5.1 \times 10^{-4}$ |
| Biosynthesis of alkaloids derived from ornithine, lysine and nicotinic acid | $9.8 \times 10^{-5}$ | $1.1 \times 10^{-3}$ |
| Biosynthesis of phenylpropanoids | $2.6 \times 10^{-4}$ | $2.5 \times 10^{-3}$ |
| Valine, leucine and isoleucine biosynthesis | $3.2 \times 10^{-4}$ | $2.9 \times 10^{-3}$ |
| Arginine and proline metabolism | $4.1 \times 10^{-4}$ | $3.4 \times 10^{-3}$ |
| Lysine biosynthesis | $6.1 \times 10^{-4}$ | $4.7 \times 10^{-3}$ |
| Biosynthesis of alkaloids derived from histidine and purine | $9.3 \times 10^{-4}$ | $6.7 \times 10^{-3}$ |
| Phenylpropanoid biosynthesis | $1.2 \times 10^{-3}$ | $8.4 \times 10^{-3}$ |
| Biotin metabolism | $1.5 \times 10^{-3}$ | $9.8 \times 10^{-3}$ |
| Cyanoamino acid metabolism | $1.9 \times 10^{-3}$ | $1.2 \times 10^{-2}$ |
| Nicotinate and nicotinamide metabolism | $2.7 \times 10^{-3}$ | $1.4 \times 10^{-2}$ |
| Pantothenate and CoA biosynthesis | $2.7 \times 10^{-3}$ | $1.4 \times 10^{-2}$ |

**Table 6.** Pathway enrichment analysis of differential metabolites in ESI− scan.

| Pathway | *p*-Value | FDR |
|---|---|---|
| Biosynthesis of unsaturated fatty acids | $2.5 \times 10^{-1}$ | $3.0 \times 10^{-9}$ |
| Alanine, aspartate and glutamate metabolism | $4.2 \times 10^{-1}$ | $2.5 \times 10^{-8}$ |
| Biosynthesis of phenylpropanoids | $5.7 \times 10^{-9}$ | $2.2 \times 10^{-7}$ |
| Aminoacyl-tRNA biosynthesis | $2.3 \times 10^{-8}$ | $5.5 \times 10^{-7}$ |
| ABC transporters | $2.1 \times 10^{-8}$ | $5.5 \times 10^{-7}$ |
| Metabolic pathways | $4.4 \times 10^{-8}$ | $8.7 \times 10^{-7}$ |
| Biosynthesis of alkaloids derived from ornithine, lysine and nicotinic acid | $7.1 \times 10^{-8}$ | $1.2 \times 10^{-6}$ |
| Arginine and proline metabolism | $4.9 \times 10^{-6}$ | $7.3 \times 10^{-5}$ |
| Biosynthesis of plant hormones | $7.6 \times 10^{-6}$ | $1.0 \times 10^{-4}$ |
| Reductive carboxylate cycle (CO2 fixation) | $1.2 \times 10^{-5}$ | $1.2 \times 10^{-4}$ |
| Glycine, serine and threonine metabolism | $5.0 \times 10^{-5}$ | $5.4 \times 10^{-4}$ |
| Pyrimidine metabolism | $1.7 \times 10^{-4}$ | $1.7 \times 10^{-3}$ |
| Glyoxylate and dicarboxylate metabolism | $2.3 \times 10^{-4}$ | $2.1 \times 10^{-3}$ |
| Two-component system | $2.6 \times 10^{-4}$ | $2.2 \times 10^{-3}$ |
| Proximal tubule bicarbonate reclamation | $3.2 \times 10^{-4}$ | $2.4 \times 10^{-3}$ |
| Phenylalanine metabolism | $3.0 \times 10^{-4}$ | $2.4 \times 10^{-3}$ |
| Biosynthesis of alkaloids derived from shikimate pathway | $4.4 \times 10^{-4}$ | $3.1 \times 10^{-3}$ |
| Citrate cycle (TCA cycle) | $6.1 \times 10^{-4}$ | $4.0 \times 10^{-3}$ |
| Biosynthesis of alkaloids derived from histidine and purine | $6.4 \times 10^{-4}$ | $4.0 \times 10^{-3}$ |
| Propanoate metabolism | $7.3 \times 10^{-4}$ | $4.3 \times 10^{-3}$ |

## 4. Discussion

For new varieties produced using genetic engineering methods, with or without the intention of modifying nutritional content, composition evaluation is used for risk assessments before commercialization and to determine whether any unintended changes of biological importance occurred that may warrant further investigation [47–50]. This ensures that genetically modified (GM) crops do not have inadvertent mutations that can lead to unintended harmful effects such as the production of toxins or toxic intermediates [47,51]. Comparison and evaluation of the composition and nutritional properties of GM crops and their wild-type counterparts are needed to determine if research in the future should position this mutant for human food.

Rice is the primary staple crop of Asia, where the average person consumes nearly 100 kg annually. While a good source of kilocalories, rice grains are considered relatively nutrient-poor because of the low levels of protein and important vitamins. This can lead to many nutritional deficiencies and diseases occurring when rice is the primary dietary component, disproportionately affecting impoverished populations that lack access to various foods. Thus, efforts have been made to biofortify rice, such as traditional and marker-assisted breeding, transgenic technology, and gene editing. An example is the famous "golden rice" strategy that produces grains enriched with a precursor for vitamin A [50,52–54]. In another study, a mutation in the DNA demethylase gene, *OsROS1*, significantly increased the vitamins and minerals compared with the wild type control [55].

Processing methods are also used to enhance the nutritional quality of threshed rice. One example is the production of rice that has been partially germinated, referred to as pre-germinated rice (PGR) and sprouted brown rice [56–58]. Pre-germinated rice and rice-based products such as multi-grain breads are available commercially. The rice germination process starts with the imbibition of water and terminates with the protrusion of the embryonic axes, either the radicle and/or the coleoptile, which usually takes place approximately 48 h after imbibition [10,59]. With faster germination rates, the *oswrky71* mutant can reduce the energy required to produce PGR, which may result in a product with more consistent end-use quality.

We examined the physical characteristics and composition of the mutant and wild-type rice kernels from the Nipponbare cultivar grown in greenhouse conditions. Due to its genome being sequenced to a high degree of accuracy, the Nipponbare cultivar has become the standard reference sequence for studying the variations among numerous rice cultivars, including their wild relatives, and other economically important cereal crops that are vital to the world's food supply [26]. A mutation in *OsWRKY71* did not affect the seed appearance and shape as both genotypes have a white seed-coat color, and the kernels are categorized as 'short' and 'bold'. Moreover, a mutation in *OsWRKY71* has no effect on 100-grain weight, one of the determinants of crop yield. To precisely determine the effect of the mutation on rice yield and any possible tradeoffs, the mutants and wild-type lines should be grown in a natural field environment, and other parameters such as tiller number, panicle size, spikelet per panicle, and grain filling percentage should be recorded [28,60].

Most proximate composition parameters reported are similar between the wild type and mutant. Although the crude protein content was significantly higher in the mutant kernels, the total protein was reduced in the mutant. This is consistent with the decrease in concentration of amino acids valine, aspartic acid, lysine, and phenylalanine. Crude protein is a measure of all nitrogen sources, including non-protein nitrogen sources [61]. Wild-type alanine, arginine, glutamic acid, and valine concentrations were higher than the literature's and the International Life Sciences Institute Crop Composition Database's (ILSI-CCDB) range for these amino acids [50,62–64]. Both wild type and *oswrky71* have higher concentrations of aspartic acid and lysine but lower methionine and tryptophan concentrations compared with the literature and ILSI-CCDB range.

Mineral analyses revealed a higher content of phosphorus and zinc in the mutant grains and lower calcium, potassium, sodium, and manganese. All minerals analyzed are within the ILSI-CCDB range except for potassium [62–64]. Both wild-type control and *oswrky71* threshed samples have potassium concentrations above the ILSI-CCDB range. Riboflavin (vitamin B2) was increased in the mutant but still within the ILSI-CCDB range. Thiamine (B1) and calcium pantothenate (B5) concentrations were reduced in *oswrky71*. Thiamine (B1) concentrations are within range in ILSI-CCDB in both wild-type and mutant kernels; however, both genotypes have slightly lower calcium pantothenate (B5) concentrations compared with the reference range. The nutrient profile of the pre-germinated mutant should still be evaluated to determine if there are differences compared with the wild type. Most comparative analyses of the nutritional composition of mutant or transgenic rice varieties were limited to comparing parameters such as proximate analyses, amino acids, minerals, vitamins, and anti-nutrient analyses [50,65–68].

With the advent of 'omics' profiling technologies, untargeted metabolomics is now being used in conjunction with current analytical methods to improve the risk assessment of genetically modified crops set by the Organization of Economic Co-operation and Development [47,49]. Compared to their wild-type counterparts, OECD defines conventional comparative methods in evaluating the safety and nutritional content of foods from GM crops using validated targeted analytical protocols [47]. These analytical methods have limitations in providing complete coverage of the metabolome changes that might be affected by genetic engineering [47]. Thus, metabolomics, which combines analytical chemistry with bioinformatics, offers a much more comprehensive and high-throughput characterization of the metabolome [47]. Proposals have been made to leverage metabolomics for safety assessment and compositional analysis of GM crops alongside the methods outlined by the OECD [47,69–71]. Interestingly, a meta-analysis on omics-profiling publications comparing GM and non-GM crops, with or without intent to change any metabolic pathway, showed that genetic modifications, specifically transgene insertion, produce few inadvertent consequences [72–74].

Comparative metabolomics analysis has been used in several rice studies designed to improve rice quality and evaluate metabolome changes that may impact the nutritional content or therapeutic potential of different rice varieties [75–78]. In the metabolic profiling of GM rice lines transformed with insect-resistance genes, the study found three upregulated metabolites in the GM lines. Still, most metabolites were equivalent to the wild-type counterpart [79]. In another study comparing GM lines with different sets of antifungal and insect-resistance genes, significant changes in protein, amino acid, vitamin, and mineral concentrations were recorded. Still, they were all within the range according to OECD values [80]. In this study, we utilized untargeted metabolic profiling and identified a few metabolites that differed between the samples in terms of individual metabolite levels, but not in terms of different metabolites. If these metabolites are shown to have the same differences between the mutant and wild type across different growing environments, they could be used as biomarkers for the *oswrky71* mutant. In addition to the results from the analyses performed in this study, the metabolic profiling provides some evidence that the *oswrky71* mutant is substantially equivalent to its wild type, which has a history of safe use. However, it should be noted that there is not a single extraction method or analytical method available, including metabolomic analysis, that can determine the entire metabolome of a plant because of the physiochemical differences between the different classes of metabolites [47].

### 5. Conclusions

In this study, we performed a comparative compositional evaluation of a rice mutant with an early germination phenotype generated via transposon-mediated knockdown of the *OsWRKY71* gene. Due to its novel germination-related phenotype, the *oswrky71* mutant can improve rice germination rates, which is advantageous for farmers in water-constrained global regions. Among the compositional parameters evaluated, the key nutritional components in the mutant rice kernel were comparable to its wild-type counterpart and within or similar to safe ranges suggested by the ILSI-CCDB for safe consumption. Further agronomic evaluation and cooked-rice sensory properties must be assessed before positioning this mutant for human consumption.

**Supplementary Materials:** The following supporting information can be downloaded at https://www.mdpi.com/article/10.3390/agronomy13041149/s1: Table S1: Identified metabolites in both WT and wrky71 under ESI+ scan; Table S2: Identified metabolites in both WT and wrky71 under ESI− scan; Table S3: Identified metabolites showing a difference between wrky71 and WT under ESI+ scan; Table S4: Identified metabolites showing a difference between wrky71 and WT under ESI− scan; Table S5: Identified metabolite pathways showing a difference between wrky71 and WT group under ESI+ scan.

**Author Contributions:** Conceptualization, S.B., Q.J.S. and C.B.; methodology, validation, formal analysis, investigation, and visualization, S.B. and A.J.V.; data curation, S.B.; writing—original draft preparation, S.B. and A.J.V.; writing—review and editing, S.B., A.J.V., Q.J.S. and C.B.; supervision, Q.J.S. and C.B.; funding acquisition, Q.J.S. and C.B. All authors have read and agreed to the published version of the manuscript.

**Funding:** This project was supported by the UNLV Faculty Opportunity Awards 2018 and a USDA grant (2018-67013-27421).

**Data Availability Statement:** Data are available from the following reference. Bataller, S. et al. (2023): Comparative Nutritional Assessment and Metabolomics of a WRKY Rice Mutant with Enhanced Germination Rates. figshare. Dataset. https://doi.org/10.6084/m9.figshare.22645096.v1. Additional data are available upon request.

**Acknowledgments:** The authors appreciate the UNLV University Libraries Open Access Funds for providing the publication fees.

**Conflicts of Interest:** The authors declare no conflict of interest. The funders had no role in the design of the study, in the collection, analyses, or interpretation of data, in the writing of the manuscript, or in the decision to publish the results.

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
