# Peer review of "Comparative Nutritional Assessment and Metabolomics of a WRKY Rice Mutant with Enhanced Germination Rates"

_agronomy, doi:10.3390/agronomy13041149_

Round 1

Reviewer 1 Report

Review remarks on the manuscript entitled Comparative nutritional assessment and metabolomics of a WRKY rice mutant with enhanced germination rates” submitted by Bataller et al. to Agronomy Journal. The era of climate change and food insecurity supports the need for rice with agronomic benefits. The assessment of major nutritional elements, compositional analysis, and comparative metabolomics on threshed grain samples was performed and assess the safety of foods from genetically modified crops. As compared to the wild type, cv. Nipponbare, oswrky71 mutant grains have similar size, shape, and amount of crude fiber, crude fat, and ash content except for higher crude protein. The higher contents of phosphorus and zinc but minimum of calcium, potassium, sodium, and manganese in the mutant. Vitamin B revealed higher riboflavin levels but minimum choline chloride, calcium pantothenate, and thiamine. Untargeted metabolomics analyses identified nearly 50 metabolites, varying between the mutant and wild types.

I found the present article very well thought through. It adequately describes every aspect from the problem background up to the conclusions.

Reviewer 2 Report

Dear Authors

The present manuscript entitled “Comparative nutritional assessment and metabolomics of a WRKY rice mutant with enhanced germination rates” discuss regarding a transposon insertional rice mutant with enhanced germination rates which may have potential to sprout under limited water conditions due to its early vigor. The manuscript is very well organized and presented; introduction includes good information while results and discussion is quite impressive. Although there are few suggestions for further improvement prior to publication, please find them here below.

Major question here is that Rice cultivation is being done with enough water availability, its major limitation. Please explain how it will be beneficial that seeds may sprout in low water availability. Even if seeds can germinate with lower water, still it will need plenty of water to complete the life cycle? 

Line 97- Please include how the seeds were germinated, in a medium or on moist filter paper?

Line 104- Please confirms if Japonica rice need 6-7 months to complete the life cycle.

Line 137- Please include which samples were used to analyze the proteins.

Line 149- Please include which samples were used.

Line 159 and 166- Please include which samples were used.

Figure 2- Please highlight the significance difference with star if any.

Please include a concise conclusion which may reflect the key message.

Please check the references style accordingly.

With Regards

Reviewer 3 Report

Revision to

Title: Comparative nutritional assessment and metabolomics of a WRKY rice mutant with enhanced germination rates

Authors: Santiago Bataller , Anne J. Villacastin , Qingxi J. Shen and Christine Bergman.

Abstract Journal: Agronomy

Manuscript number: agronomy-2297087

General remarks: The manuscript by Bataller et al., showed an interesting characterization about a rice mutant genotype modified in an WRKY gene. The authors focused on morphological, qualitative and metabolomics aspects of rice seeds comparing a Wt genotype and this mutant genotype. This manuscript need to be extensively revised before acceptance. Particularly, the authors should improve the description of the results, highlighting the best results and giving more importance to novelty of the manuscript. In my opinion the most interesting analysis of the manuscript is the metabolomics but the authors did not described well these experiments. Very few and too generic information were described to metabolites differentially produced in the two genotypes. Figures are not adequate to show these results and the tables were confined in the supplemental. Furthermore, the manuscript is not properly written in English language showing grammar errors and poor fluency. The authors should significantly improve this aspect. Based on these evaluations, this manuscript will be considered for acceptance on Agronomy after substantial revisions.

Abstract: the abstract is too schematic showing poor fluency. In some part, this section appears a list of separate sentences. Please reorganize the entire abstract.

Introduction:

Line 30: I do not understand why the authors use “is expected to remain”. Simply, Rice is an essential an essential source….

Lines 43-48. Please improve this part of the introduction Authors should detail the effects of climate change (and abiotic stresses) on Rice cultivation specifying the effects on rice cultivation (maybe in the aspects analyzed by the authors in main text), and citing numbers about the effects of the climate change on rice production.

Line 69: It is not clear when the authors identified this rice mutant line. This is the first  manuscript in which the authors describe this? There are other studies on this specific mutants by the authors or by others? Please give more details.

Line 76: “OsWRKY71 has been previously shown to be involved in the complex interaction of GA and ABA during germination” – please add a reference.

Methods:

Line 95: is this a reference “(Kumar, Wing, and Sundaresan 2005)” ? In this case authors should cite this following the MDPI guidelines.

Line 103: Please add a specific paragraph about DNA extraction and PCR methods

Results:

Line 204: are the authors sure that mean “barley”?

Lines 204-213: this part is too descriptive for the results paragraph, please move these information in other section. 

Line 221: this is a method, please move this sentence in the appropriate paragraph. Please avoid the repetition of methods information in the results.

Paragraphs 3.3, 3.4, 3.5: Please give more quantitative and qualitative details about the significant results showed in the results section. This part of the manuscript should be improved giving to the reader more information about the most interesting results.

Paragraph 3.6 should be reduced and merged with 3.7. It is not necessary explain metabolomics method in the results section.

Paragraph 3.7: see general remarks

Discussion: Discussion is confusingly written. This section should extensively revised. The authors should also focused on most interesting differences of their genotypes comparing their data with similar evaluations in literature.

Figures

Figure 2A showed two totally black boxes, please correct this problem.

Table 1: please highlight the significant differences in the table.

Figure 3 is less informative. I suggest to Authors to move this figure into supplemental.

Round 2

Reviewer 3 Report

The R1 version of the manuscript by Battaler is significantly improved. The authors did not answer to any revision points but reported acceptable justifications. I do not fully agree with the authors but the but overall the manuscript is acceptable for publication. Particularly, some corrections made the manuscript more precise, improving the quality of presentation and English fluency. Based on these considerations, this manuscript  is accepted for publication.